# Oligo-FISH of *Populus simonii* Pachytene Chromosomes Improves Karyotyping and Genome Assembly

**DOI:** 10.3390/ijms24129950

**Published:** 2023-06-09

**Authors:** Yilian Zhao, Guangxin Liu, Ziyue Wang, Yihang Ning, Runxin Ni, Mengli Xi

**Affiliations:** State Key Laboratory of Tree Genetics and Breeding, Co-Innovation Center for Sustainable Forestry in Southern China, College of Forestry, Nanjing Forestry University, Nanjing 210037, China; yilianzhao@njfu.edu.cn (Y.Z.); guangxinliu@njfu.com.cn (G.L.); zyw@njfu.edu.cn (Z.W.); ningyihang@njfu.edu.cn (Y.N.);

**Keywords:** pachytene chromosome, oligo-FISH, karyotype, repeats assembly, *Populus simonii*, forest model species

## Abstract

Poplar was one of the first woody species whose individual chromosomes could be identified using chromosome specific painting probes. Nevertheless, high-resolution karyotype construction remains a challenge. Here, we developed a karyotype based on the meiotic pachytene chromosome of *Populus simonii* which is a Chinese native species with many excellent traits. This karyotype was anchored by oligonucleotide (oligo)-based chromosome specific painting probes, a centromere-specific repeat (Ps34), ribosomal DNA, and telomeric DNA. We updated the known karyotype formula for *P. simonii* to 2n = 2x = 38 = 26m + 8st + 4t and the karyotype was 2C. The fluorescence in situ hybridization (FISH) results revealed some errors in the current *P. simonii* genome assembly. The 45S rDNA loci were located at the end of the short arm of chromosomes 8 and 14 by FISH. However, they were assembled on pseudochromosomes 8 and 15. In addition, the Ps34 loci were distributed in every centromere of the *P. simonii* chromosome in the FISH results, but they were only found to be present in pseudochromosomes 1, 3, 6, 10, 16, 17, 18, and 19. Our results reveal that pachytene chromosomes oligo-FISH is a powerful tool for constructing high-resolution karyotypes and improving the quality of genome assembly.

## 1. Introduction

The karyotype describes the number and appearance of all chromosomes and provides the most basic genomic information about a eukaryotic species [1]. Chromosomes of mitotic metaphase or meiotic pachytene can be used for karyotype analysis. Metaphase chromosomes are suitable for the karyotype analysis of species with large chromosomes or obvious chromosomal features. Chromatin at the pachytene stage is not completely compressed and the length is 10–50 times that in metaphase chromosomes [2]. This means that pachytene chromosomes are more suitable for karyotype analysis than metaphase chromosomes, particularly for species with small and uniform size chromosomes [3]. Accurate identification of each chromosome is the basis of a karyotype analysis. Chromosome identification techniques have been developed from traditional chromosome morphology observation, and chromosome banding technology to fluorescence in situ hybridization (FISH) [4,5]. The application of FISH has greatly improved research in the field of karyotype analysis [6]. The probes for FISH have evolved from rDNA, bacterial artificial chromosomes (BACs) to highly specific oligonucleotides (oligos). At present, oligo-FISH can distinguish both non-homologous chromosomes and homologous chromosomes [7]. Oligo-FISH has been used for plant genome evolution tracing [5], the study of chromosome behavior [8], and to improve genome sequence assembly [9]. It has become a powerful tool for molecular cytogenetics and genomics research.

Poplar is a dioecious tree and has become a model species in forest research due to its biological characteristics and its economic and ecological value. However, poplar chromosomes are small and similar in size, which means that it is difficult to identify each chromosome accurately using traditional cytogenetic methods [5]. In recent years, the rapid development of plant molecular cytogenetics technologies, especially the creation and application of oligo-FISH technology in plants [5], has expanded molecular cytogenetics research into poplar. We previously developed a complete set of 19 chromosomes painting probes based on the reference genome of *Populus trichocarpa*. The results showed that the probes were universal for *Populus* species from five different sections [1]. We demonstrated various applications of the chromosome painting probes in genetics and genomic studies in poplar. In addition, we found a pairing delay at the distal ends of the sex chromosome during the pachytene stage [9]. We then used sequential oligo-FISH to construct metaphase chromosome karyotypes for *P. trichocarpa* and *P. euphratica* [1]. These studies have promoted research on molecular cytogenetics of poplar, but there are still many questions that need to be answered. For example, although the 5S rDNA was located on chromosome 17 by FISH, it still could not confirm whether the site was on the long or short arm.

*Populus simonii* Carr. belongs to the genus *Populus* in the Salicaceae family. It is a tree species that is native to China, has a wide distribution, is drought resistant, tolerant of barren land, has wide adaptability, and has a strong rooting ability. Therefore, *P. simonii* is one of the best parents for poplar breeding [10]. *P. simonii* genome sequencing and assembly results, based on PacBio long-read sequencing, were published by Wu et al. [11]. Lan et al. reported the meiosis process of *P. simonii* pollen mother cells [12], which provided a reference for collecting the male buds at the pachytene stage. In this study, chromosome-specific oligos, a centromere-specific repeat (Ps34), rDNA, and telomeric DNA were labeled as the FISH probes respectively. These probes were hybridized with *P. simonii* chromosomes. Based on the accurate identification of each chromosome and the centromere location, a high-resolution pachytene karyotype was constructed. Interestingly, the 45S rDNA were assembled on pseudochromosomes 8 and 15 in the currently published genome, but their FISH signals suggested that they were located on chromosomes 8 and 14. Although Ps34 repeats were detected in every centromere region of the *P. simonii* chromosome by FISH, they were only assembled in 8 of the 19 pseudochromosomes. This study provided a solid foundation for research on molecular cytogenetics of small chromosome species and further improved the assembling quality of current *P. simonii* genome.

## 2. Results

### 2.1. Karyotyping of Metaphase Chromosomes

Chromosome painting probes for all 19 chromosomes were developed based on the reference genome of *P. trichocarpa* [1]. We first examined the validity of these probes in *P. simonii*. Each of the 19 chromosome painting probes generated bright signals on its corresponding metaphase chromosome. A centromeric repeat probe and 18 chromosome painting probes were used in the karyotype analysis. By sequential FISH, we identified all chromosomes in the same metaphase cells and then measured the length of the short and long arms of each chromosome. We used seven rounds of sequential FISH to distinguish all 19 pairs of chromosomes (Figure 1h). In each of the first six rounds, three different painting probes (digoxigenin-labelled, biotin-labelled, and labelled by both digoxigenin and biotin) were hybridized into three chromosomes (Figure 1a–f). Ps34 was used in the final round to indicate the positions of the centromeres of all the chromosomes (Figure 1g). The measurement was conducted on each chromosome in 10 complete metaphase cells and the data is shown in Table 1. *P. simonii* is diploid with 19 pairs of homologous chromosomes (2n = 2x = 38). However, due to the small and uniform size of most of the chromosomes, it was difficult to construct a credible karyotype based on the metaphase chromosomes.

### 2.2. Comparing the Three Methods for Pachytene Chromosomes Preparation

Three methods, squashing, smearing, and dropping, were used to prepare the pachytene chromosomes (see Section 4). The effectiveness of these methods was compared by detecting chromosome 19 using FISH. Previous studies have shown that the distal region of the short arm of chromosome 19, which accounts for ~8% of the chromosome, was not labeled by the available chromosome painting probes [9]. Thus, the long and short arms of chromosome 19 can be distinguished by their signal patterns. We measured the long and short arm lengths of pachytene chromosome 19 that had been prepared by squashing and found that the short arms were longer than the long arms in three of the 10 cells (Figure 2a). This suggested that local mechanical stretching had occurred. In addition, broken chromosome 19 was frequently observed on the slides (Figure 2b). Pachytene chromosomes from one cell always tangled together when they were prepared by smearing (Figure 2d) and this made them difficult to analyze. When we selected the dropping method to prepare the pachytene chromosomes, we found each short arm was shorter than the corresponding long arm (Figure 2c). Overall, the results showed that the dropping method was the most suitable for pachytene chromosome preparation.

### 2.3. Karyotyping of Pachytene Chromosomes

Chromosomes at the pachytene stage have a much higher resolution than those at the metaphase stage and are usually used to construct the standard karyotype for plant species with small chromosomes [13]. In this study, FISH was performed on the pachytene chromosomes using the painting probes, a centromeric DNA probe, a telomeric DNA probe, and rDNA probes. The results showed that the 5S rDNA was located in the middle of the short arm of chromosome 17 and the 45S rDNA located at the end of the short arms of chromosomes 8 and 14. Notably, all the 45S rDNA signals were tangled together. Surprisingly, two regions on chromosome 8 were consistently not labeled by the painting probes (Figure 3). One region was on the long arm near the centromere and the other was on the short arm and was next to the 45S rDNA, which accounted for about 8% of the total length of chromosome 8 (excluding 45S rDNA). There were three signals for the telomere probe on chromosome 10. Two were located at both ends of chromosome 10 and another one was located on the short arm (Figure 3 Yellow arrow). This phenomenon has also been reported in barley [14]. The length of each chromosome arm was measured in 30 pachytene chromosomes (Table 1). The ideogram of the karyotype (Figure 4) was completed according to the data of pachytene chromosomes in Table 1. The pachytene chromosomes were much longer than metaphase chromosomes and their lengths varied from 8.66 (Chr. 13) to 52.92 (Chr. 1) μm, whereas those at the metaphase stage varied from 0.69 (Chr. 7) to 1.28 (Chr. 1) μm. Among them, 13 chromosomes (Chr. 1, 4, 5, 6, 7, 11, 12, 13, 15, 16, 17, 18, and 19) were metacentric chromosomes, 4 chromosomes (Chr. 2, 3, 8, and 10) were acrocentric chromosomes, and 2 chromosomes (Chr. 9 and Chr. 14) were telocentric chromosomes. The karyotype formula was 2n = 2x = 26m + 8st + 4t.

## 3. Discussion

The karyotype of *P. simonii* has already been reported by Chen et al. [15]. They measured the short and long arms of chromosomes taken from *P. simonii* root tips. Based on Li’s criterion [16], they developed a *P. simonii* karyotype formula, which was 2n = 2x = 38 = 1M + 26m + 8sm + 1st + 2t and the karyotype type was 2B [15]. In this study, oligo-based chromosome specific painting probes and centromere-specific repeat probes were used to mark the chromosomes and centromeres. Furthermore, sequential FISH was able to identify all the chromosomes and the location of the centromeres in the same metaphase cells. According to Li’s criterion, the karyotype formula of *P. simonii* is 2n = 2x = 38 = 26m + 10sm + 2st and the karyotype type is 2C. Although the centromere positions were successfully marked, measurement error was inevitable due to the small chromosome size. In order to further improve the resolution, we developed a karyotype based on meiotic pachytene chromosomes from anthers of *P. simonii*. The karyotype was anchored by oligo-based chromosome specific painting probes, a centromere-specific repeat, rDNA, and telomeric DNA. According to Li’s criterion, the karyotype formula of *P. simonii* is 2n = 2x = 38 = 26m + 8st + 4t, which belongs to 2C. The metaphase chromosome karyotype of *P. simonii* published by Chen et al. [15] is different from the one produced by this study. This may be because the method used by Chen et al. did not accurately identify each chromosome and the centromere positions. There were also some differences between the karyotypes produced from the metaphase chromosomes and the pachytene chromosomes in this study. This may be due to errors when measuring the metaphase chromosomes. Therefore, for species with small chromosomes, it is recommended that pachytene chromosomes are used to construct karyotypes.

The arm ratio is the basis for chromosome classification. Therefore, obtaining the accurate length of the long and short arms of chromosomes is the prerequisite for constructing high-quality karyotypes [17]. The chromosome lengths at the pachytene stage were 10–50 times that at the metaphase stage and therefore pachytene chromosomes were more suitable for high-resolution karyotype construction [2]. However, pachytene chromosomes are not fully compressed and always intertwine with each other during chromosome preparation. In order to obtain a more dispersed pachytene chromosome, profile a squashing method is usually used [18]. However, slight slithering of the coverslip during squashing will result in mechanical stretching of the chromosome local region [19]. In this study, it was also difficult to obtain *P. simonii* pachytene chromosomes that were well dispersed using the squashing method. Dispersion can be improved by increasing the strength of squashing, but local mechanical stretching of the pachytene chromosome will increase. Twenty-six of the 38 chromosomes in *P. simonii* are metacentric chromosomes. In particular, the arm ratios of chromosomes 4, 11, and 19 were all less than 1.2, which meant that even slight local mechanical stretching affected karyotype construction. Our results showed that the dropping method maintained the original state of pachytene chromosomes. Therefore, this study selected the dropping method to prepare the *P. simonii* pachytene chromosomes. During pachytene chromosome karyotyping, the absolute length of each chromosome was measured from 30 different cells. The pachytene stage is a dynamic process, and the chromosome compression degree varies. For example, chromosome 2 was compressed by about 33.5 times [(23.69 + 6.42)/(0.53 + 0.37) ≈ 33.5] from the pachytene stage to the metphase stage, while chromosome 3 was only compressed by about 14.7 times [(9.66 + 3.01)/(0.56 + 0.3) ≈ 14.7] (Table 1). Therefore, the karyotype constructed in this study cannot accurately present the absolute length difference among chromosomes.

The construction of a karyotype lays a foundation for the study of karyotype evolution in *Populus* species and can improve the quality of the assembled genome. Highly repetitive sequences have always been a barrier to genome assembly [20]. The assembly quality of these repetitive sequences has been improved with the development of sequencing techniques. However, centromere and rDNA assembly are still very difficult [21]. The FISH technique can therefore assist in genome assembly [9]. The genome sequencing and assembly results of *P. simonii* based on PacBio long-read sequencing were published in 2020 [11]. We tested the assembly of rDNA and typical centromeric repeat sequence Ps34 in the genome and found that 5S rDNA was assembled in the middle of the short arm of pseudochromosome 17, which was consistent with the FISH results produced by this study. This indicated that 5S rDNA was assembled correctly in the *P. simonii* genome. Interestingly, 45S rDNA was assembled on pseudochromosomes 8 and 15, but the FISH results showed that the 45S rDNA was located at the short arm ends of chromosomes 8 and 14. Surprisingly, Ps34 was only present in *P. simonii* pseudochromosomes 1, 3, 6, 10, 16, 17, 18, and 19. However, the FISH results showed that the Ps34 loci were distributed in the centromeric region of every *P. simonii* chromosome (Figure 3). This meant there were still some mistakes in 45S rDNA and centromere region assembly in *P. simonii*. In addition, a large number of contigs were not anchored on the *P. simonii* pseudochromosomes. In the future, we could select unique sequences for these contigs. These sequences will then be anchored directly into chromosomes using FISH [1].

The chromosome specific painting probes used in this study were single-copy oligos from each of the 19 pseudochromosomes of the *P. trichocarpa* genome. These oligos covered the entire pseudochromosome as much as possible [1]. Using this set of 19 chromosome painting probes, no inter-chromosomal translocations were observed at the cytological level among five poplar species belonging to five different sections [1]. These probes also cannot be used to detect intra-chromosomal inversion because oligos from one entire pseudochromosome were mixed together. With the development of oligo-FISH technology, the design of oligo probes is becoming more flexible [22]. In the future, an oligo probe library could be designed according to specific experimental purposes. If the purpose is only to identify each chromosome, oligos located in specific regions can be selected and labeled as bar codes for each chromosome, such as has been done with a potato [23]. Based on these bar codes, each chromosome can be distinguished by a single round of FISH. In addition, oligo probes can be used to detect rearrangement among related species if they were designed and synthesized according to specific chromosome segments [24,25]. For example, chromosome 19 has been consistently reported to be involved in sex determination in *Populus* species [26,27,28,29,30,31,32]. The sex-determining locus has been mapped to different regions of the chromosome 19 short arm among different species [26,27,28,29,31,32,33]. If the short arm of chromosome 19 is divided into 2–4 segments, then oligo probes for each part can be designed and synthesized and possible rearrangements in the short arm of chromosome 19 among different *Populus* species can be revealed by FISH.

## 4. Materials and Methods

### 4.1. Plant Materials

Young *P. simonii* root tips were excised when they had reached 1–2 cm in length and were treated with nitrous oxide (N_2_O) gas at a pressure of 160 psi (~10.9 atm) for 2 h. Then, the root tips were fixed in Carnoy’s solution (acetic acid: ethanol = 1:3, *v/v*) and stored at –20 °C until use. *P. simonii* male flower buds with anthers at the pachytene stage were collected and fixed with freshly prepared Carnoy’s solution according to Xin et al. [18]. They were then stored at −20 °C.

### 4.2. Mitotic Metaphase Chromosome Preparation

A modified smear method was used to prepare mitotic metaphase chromosomes [1]. About 15 fixed root tips were rinsed in distilled water and digested with 40 μL mixed enzyme solution containing 4% cellulase, 2% pectinase, and 1% pectolyase dissolved in 0.01 M citrate buffer (pH 4.5) at 37 °C for 1 h. The softened material was then carefully rinsed in water and transferred to a slide. Meristem cells were released with tweezers and macerated in 20 μL of 60% acetic acid. Then, the slide was placed on a heater (HI 1220, Leica, Wetzlar, Germany) at 55 °C, the sample solution was smeared with a needle, washed immediately with Carnoy’s fixative, and then air dried. The slides were screened under a phase contrast microscope and those with well-spread mitotic chromosome preparations were selected for FISH.

### 4.3. Meiotic Pachytene Chromosome Preparation

Three pachytene chromosome preparation methods were compared. The first was a modified squashing method [18]; the second was the smearing method used in the metaphase chromosome preparation described above; and the third was a modified dropping method. The fixed *P. simonii* inflorescences were rinsed in distilled water and the anthers were removed from the florets. About 150 anthers were put into 100 μL mixed enzyme solution, which was the same as that used for the root tips (See Section 4.2). The mixture was digested at 37 °C for 2 h and then mixed using a pipette. The mixture was then centrifuged at 8000 rpm for 1 min and the supernatant was discarded. The remaining substances were rinsed once with 100 μL distilled water and twice with absolute alcohol. Finally, the pellet was resuspended in 100 μL Carnoy’s solution. Then, aliquots (8 μL) of the resulting suspension were dropped from a height of 5–15 cm onto a slide [34]. The slide was air-dried at room temperature and placed in a 37 °C incubator for 3 h to attach the chromosomes firmly to the slide.

### 4.4. Probes

The plasmids containing 5S and 45S rDNAs were kindly provided by Professor Jiming Jiang (Michigan State University, East Lansing, MI, USA). A centromeric repeat Ps34 was amplified from *P. simonii* (Appendix A). The 5S and 45S rDNAs were cloned from rice (*Oryza sativa*). A plasmid containing (TTTAGGG)n sequences was used to label the telomeric ends of *P. simonii* chromosomes. All the repeats were labeled with digoxigenin-dUTP (digoxigenin-11-dUTP, 11093088910, Roche, Basel, Switzerland) or biotin-dUTP (biotin-16-dUTP, 11093070910, Roche, Basel, Switzerland), using nick translation. Chromosome-specific oligos were developed based on the reference genome of *P. trichocarpa* [1,35]. The oligo probes were prepared according to Xin et al. [1].

### 4.5. Fluorescence In Situ Hybridization

FISH was performed according to published protocols with minor modifications [1]. A centromeric repeat probe and chromosome specific painting probes were used to distinguish the metaphase chromosomes. The hybridization mixture (50% formamide, 10% dextran sulfate, 2× SSC, and 100 ng chromosome painting probes) was applied to a denatured chromosomal slide, which was then incubated for 24 h at 37 °C. High-quality slides were used for sequential FISH, as described previously [1]. The centromeric repeat probe was used during the last round of FISH.

For pachytene chromosome FISH, the hybridization mixture (50% formamide, 10% dextran sulfate, 2× SSC, 100 ng chromosome painting probes, 40 ng telomere probe, 40 ng centromeric repeat probe, and 40 ng 45S and 5S rDNA probes for chromosomes 8 and 14) was applied to a denatured chromosomal slide and incubated for 24 h at 37 °C. The digoxigenin-labelled probes were detected by rhodamine anti digoxigenin (Anti-Digoxigenin-Rhodamine Fab fragments, 11207750910, Roche, Basel, Switzerland), and the biotin-labelled probes were detected using Alexa Fluor 488 streptavidin (Alexa Fluor 488 streptavidin, S11223, Invitrogen, Waltham, MA, USA). The chromosomes were counter-stained with DAPI in VectaShield Antifade solution (Vector Laboratories, Newark, CA, USA). FISH images were captured using a CCD camera (C11440-42U, Hamamatsu, Shizuoka, Japan) attached to an Olympus BX51 fluorescence microscope (Olympus, Tokyo, Japan). The final image contrast was processed using Adobe Photoshop 5.0.

### 4.6. Cytological Measurements and Analysis

The short (S) and long (L) arms of the individual chromosomes (not including the 45S rDNA region) were measured using the “measurement” tool of HC Image Live (Olympus, Tokyo, Japan). For karyotype construction, the chromosomes from 10 complete metaphase cells and the pachytene chromosomes from 30 meiosis cells were measured and standard deviations were calculated.

## 5. Conclusions

We identified all *P. simonii* chromosomes in the same metaphase cells using sequential FISH. Due to the small and uniform size of most of the chromosomes, it was difficult to construct a credible karyotype based on the metaphase chromosomes. Considering that chromosomes at the pachytene stage have a much higher resolution than those at the metaphase stage, we prepared the chromosomes using pachytene stage cells. We found the dropping method was more suitable than squashing and smearing methods for pachytene chromosome preparation. We updated the known karyotype formula for *P. simonii* based on pachytene chromosome analysis. Our results revealed that the 5S rDNA was assembled correctly, but 45S rDNA and centromere region were assembled mistakenly in the current reference genome of *P. simonii*. In conclusion, pachytene chromosomes oligo-FISH is a powerful tool for constructing high-resolution karyotypes and improving the quality of genome assembly.

## Figures and Tables

**Figure 1 ijms-24-09950-f001:**
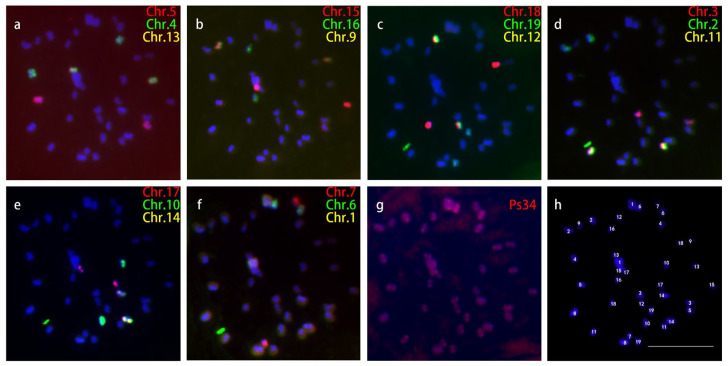
Sequential FISH to identify all 19 chromosomes in the same metaphase cell of *P. simonii*. (**a**) The first round of FISH uses three chromosome-specific painting probes, red for chromosome 5, green for chromosome 4, and yellow for chromosome 13. (**b**) The second round of FISH using three chromosome-specific painting probes, red for chromosome 15, green for chromosome 16 and yellow for chromosome 9. (**c**) Third round of FISH using chromosome-specific painting probes, red for chromosome 18, green for chromosome 19 and yellow for chromosome 12. (**d**) The fourth round of FISH using chromosome-specific painting probes, red for chromosome 3, green for chromosome 2 and yellow for chromosome 11. (**e**) The fifth round of FISH using chromosome-specific painting probes, red for chromosome 17, green for chromosome 10 and yellow for chromosome 14. (**f**) The sixth round of FISH using chromosome-specific painting probes, red for chromosome 7, green for chromosome 6 and yellow for chromosome 1. (**g**) Seventh round of FISH using centromere-specific probe Ps34 (red). (**h**) Chromosomes identified based on the above six round FISH results. Bars, 10 μm.

**Figure 2 ijms-24-09950-f002:**
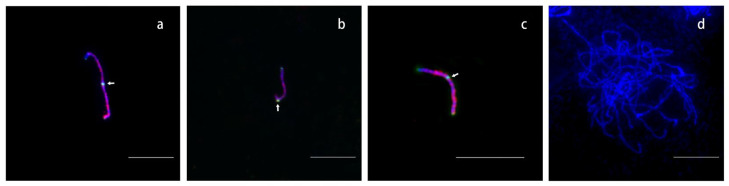
Pachytene chromosomes prepared by three methods. The painting probe of chromosome 19 is detected by red color. The telomere and centromere probes are detected by green color. White arrows point to centromere signal sites. (**a**,**b**) Pachytene chromosome 19 prepared by squashing. a, showing short arm is longer than long arm. b, showing the chromosome is broken. (**c**) Pachytene chromosome 19 prepared by dropping. (**d**) Pachytene chromosomes prepared by smearing. Bars, 10 μm.

**Figure 3 ijms-24-09950-f003:**
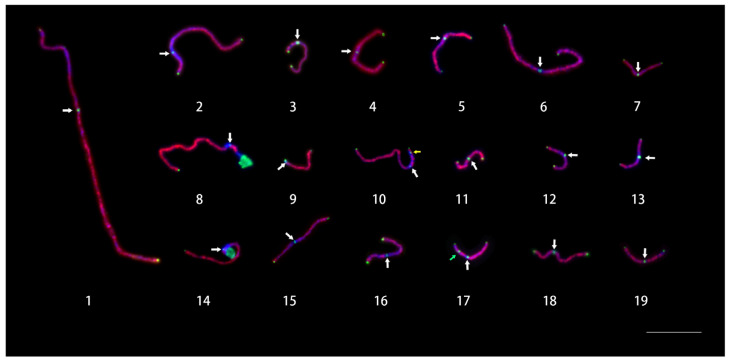
19 pachytene bivalents of *P. simonii*. were identified by FISH. The chromosomes painting probes are detected by red color. White arrows point to centromere signal sites. The telomeres of all chromosomes are detected by green color at the ends of chromosomes. 45S rDNA are detected as a bunch of green signals at the end of short arms of chromosome 8 and chromosome 14. The green arrow points to the signal of 5S rDNA on the short arm of chromosome 17. The yellow arrow points to the signal of the telomere probe on the short arm of chromosome 10. Bars, 10 μm.

**Figure 4 ijms-24-09950-f004:**
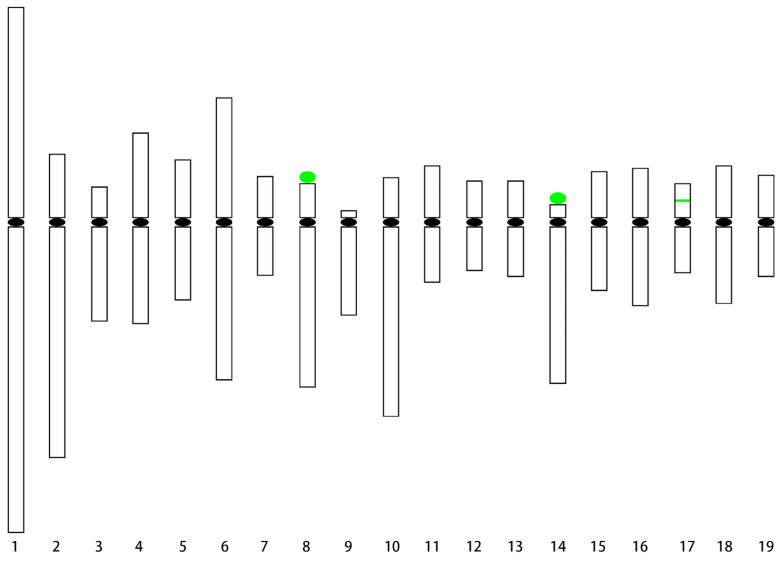
Ideogram of the karyotype of *P. simonii*. Note: black ellipses indicate the centromere position, green ellipses indicate the 45S rDNA position, and green strip indicate the 5S rDNA position.

**Table 1 ijms-24-09950-t001:** Lengths of long and short arms and arm ratios of *P. simonii* chromosomes.

Chr. No.	Long Arm (µm)		Short Arm (µm)		Arm Ratio ^c^	
	Metaphase ^a^	Pachytene ^b^	Metaphase ^a^	Pachytene ^b^	Metaphase	Pachytene
1	0.76 ± 0.11	31.46 ± 9.21	0.52 ± 0.10	21.46 ± 6.62	1.46 ± 0.17	1.48 ± 0.11
2	0.53 ± 0.08	23.69 ± 4.45	0.37 ± 0.04	6.42 ± 1.17	1.46 ± 0.16	3.69 ± 0.18
3	0.56 ± 0.09	9.66 ± 0.69	0.30 ± 0.03	3.01 ± 0.31	1.90 ± 0.31	3.23 ± 0.28
4	0.47 ± 0.02	10.02 ± 2.260	0.33 ± 0.03	8.40 ± 1.67	1.43 ± 0.16	1.19 ± 0.07
5	0.48 ± 0.05	7.62 ± 0.69	0.36 ± 0.04	5.83 ± 0.45	1.35 ± 0.14	1.31 ± 0.09
6	0.48 ± 0.05	15.85 ± 2.80	0.37 ± 0.05	12.27 ± 2.13	1.33 ± 0.19	1.29 ± 0.07
7	0.39 ± 0.05	4.95 ± 0.44	0.30 ± 0.04	4.03 ± 0.38	1.28 ± 0.09	1.23 ± 0.08
8 ^d^	0.55 ± 0.07	16.37 ± 2.33	0.24 ± 0.04	3.33 ± 0.49	2.34 ± 0.27	4.93 ± 0.32
9	0.54 ± 0.05	9.05 ± 0.94	0.18 ± 0.03	0.52 ± 0.12	3.09 ± 0.59	18.21 ± 5.04
10	0.58 ± 0.09	19.53 ± 5.86	0.32 ± 0.06	3.83 ± 0.81	1.85 ± 0.33	5.06 ± 0.87
11	0.53 ± 0.09	5.49 ± 0.70	0.37 ± 0.04	5.00 ± 0.61	1.48 ± 0.20	1.10 ± 0.04
12	0.41 ± 0.04	4.38 ± 0.50	0.31 ± 0.04	3.61 ± 0.38	1.31 ± 0.14	1.21 ± 0.08
13	0.47 ± 0.03	5.06 ± 0.49	0.34 ± 0.06	3.60 ± 0.42	1.41 ± 0.26	1.41 ± 0.11
14 ^d^	0.63 ± 0.07	15.92 ± 2.00	0.23 ± 0.04	1.14 ± 0.29	2.75 ± 0.45	14.81 ± 3.85
15	0.46 ± 0.03	6.57 ± 0.96	0.30 ± 0.04	4.35 ± 0.67	1.54 ± 0.24	1.52 ± 0.08
16	0.49 ± 0.06	8.00 ± 1.51	0.29 ± 0.05	4.70 ± 0.91	1.74 ± 0.31	1.70 ± 0.06
17	0.44 ± 0.07	4.70 ± 0.39	0.31 ± 0.03	3.28 ± 0.311	1.39 ± 0.20	1.43 ± 0.06
18	0.46 ± 0.04	7.76 ± 1.16	0.34 ± 0.03	5.24 ± 0.74	1.39 ± 0.18	1.48 ± 0.09
19	0.42 ± 0.06	5.06 ± 0.62	0.35 ± 0.05	4.27 ± 0.51	1.22 ± 0.14	1.19 ± 0.06

^a^ Average length of each chromosomal short and long arms from 10 metaphase cells. ^b^ Average length of each chromosomal short and long arms from 30 meiosis cells. ^c^ Arm ratio = length of the long arm/length of the short arm. ^d^ The 45S rDNA on the short arm of chromosome 8, 14 were not included in the measurement.

## Data Availability

The data presented in this study are available on request from the corresponding author.

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
