# Peer review of "Oligo-FISH of Populus simonii Pachytene Chromosomes Improves Karyotyping and Genome Assembly"

_ijms, 2023, doi:10.3390/ijms24129950_

Round 1

Reviewer 1 Report

The paper by Dr Zhoa and coauthors describes a FISH method for karyotyping the Asian aspen species Populus simonii using pachytene phase chromosomes. Results from this work were used to assess a previously published genome for P. simonii based on long read sequence data. The main innovations presented in this paper were the optimization of slide mounting methods and the use of pachytene phase tissue. The authors provide an apparently improved karyotype for this species however differences in relative lengths between the assembled long read genome and their karyotype as presented in figure 4 should be explained. For instance, according to the long read genome, chromosomes 2 (24.81 Mb) and 3 (23 Mb) should be nearly the same length but differ greatly in the karyotype i.e. chr 2 is twice as long as 3 in the karyotype. Could these differences be the result of mislabeling? Or the result of assembly errors in repetitive regions? Or damage to the pachytene chromosomes in slide preparation? The authors need to discuss this in greater detail. Also, as a result of these differences could the conclusions regarding the discrepancies between 45S placement (chr 14 in the karyotype and chr 15 in the long read genome) in the long read genome and the karyotype be incorrect? This should also be discussed in greater detail.

Numerous grammatical errors and unclear phrasing were found in the manuscript. This reviewer made a cursory pass in correcting errors (see attached comments in the manuscript) but a greater attention to proofreading should be made in the revised version.

Reviewer 2 Report

I checked your manuscript and described comments below.

Populus simonii is a tree distributed in China, Europe and North America.

In this paper, high-resolution karyotype analysis using the Origo-FISH method is performed for Populus simonii.

I have a questions and comments as follows.

1.       I think it would be better to add a table of the DNA sequence for the FISH probe.

2.       Using Adobe Photoshop to correct the image of the experimental data makes it possible to create beautiful results. However, corrections are usually made with the software that comes with the microscope. If possible, I think that it is better to use the image corrected with the software attached to the microscope.

3.       I think you should put Populus simonii in the title instead of poplar.

I don't think this paper has any major mistakes or grammatical problems.
